# Characterizing Risk Factors for Hospitalization and Clinical Characteristics in a Cohort of COVID-19 Patients Enrolled in the GENCOV Study

**DOI:** 10.3390/v15081764

**Published:** 2023-08-18

**Authors:** Gregory Morgan, Selina Casalino, Sunakshi Chowdhary, Erika Frangione, Chun Yiu Jordan Fung, Simona Haller, Elisa Lapadula, Mackenzie Scott, Dawit Wolday, Juliet Young, Saranya Arnoldo, Navneet Aujla, Erin Bearss, Alexandra Binnie, Yvonne Bombard, Bjug Borgundvaag, Laurent Briollais, Marc Dagher, Luke Devine, Hanna Faghfoury, Steven M. Friedman, Anne-Claude Gingras, Lee W. Goneau, Zeeshan Khan, Tony Mazzulli, Shelley L. McLeod, Romina Nomigolzar, Abdul Noor, Trevor J. Pugh, David Richardson, Harpreet Kaur Satnam Singh, Jared Simpson, Seth Stern, Lisa Strug, Ahmed Taher, Jordan Lerner-Ellis, Jennifer Taher

**Affiliations:** 1Lunenfeld-Tanenbaum Research Institute, Toronto, ON M5G 1X5, Canada; 2Pathology and Laboratory Medicine, Mount Sinai Hospital, Toronto, ON M5G 1X5, Canada; 3Department of Laboratory Medicine and Pathobiology, University of Toronto, Toronto, ON M5S 1A8, Canada; 4William Osler Health System, Brampton, ON L6R 3J7, Canada; 5Mount Sinai Academic Family Health Team, Mount Sinai Hospital, Toronto, ON M5T 3L9, Canada; 6Department of Family and Community Medicine, University of Toronto, Toronto, ON M5G 1V7, Canada; 7Department of Critical Care, William Osler Health System, Etobicoke, ON M9V 1R8, Canada; 8Li Ka Shing Knowledge Institute, St. Michael’s Hospital, Toronto, ON M5B 1A6, Canada; 9Institute of Health Policy, Management and Evaluation, University of Toronto, Toronto, ON M5T 3M6, Canada; 10Schwartz/Reisman Emergency Medicine Institute, Sinai Health System, Toronto, ON M5G 2A2, Canada; 11Women’s College Hospital, Toronto, ON M5S 1B2, Canada; 12Division of General Internal Medicine, Mount Sinai Hospital, Toronto, ON M5G 1X5, Canada; 13Faculty of Medicine, University of Toronto, Toronto, ON M5S 1A8, Canada; 14Fred A Litwin Family Centre in Genetic Medicine, University Health Network & Mount Sinai Hospital, Toronto, ON M5T 3H7, Canada; 15Emergency Medicine, University Health Network, Toronto, ON M5G 2C4, Canada; 16Department of Molecular Genetics, University of Toronto, Toronto, ON M5S 1A8, Canada; 17Dynacare Medical Laboratories, Brampton, ON L6T 5V1, Canada; 18Mackenzie Health, Richmond Hill, ON L4C 4Z3, Canada; 19Department of Microbiology, Mount Sinai Hospital, Sinai Health, Toronto, ON M5G 1X5, Canada; 20Princess Margaret Cancer Centre, University Health Network, Toronto, ON M5G 2C4, Canada; 21Ontario Institute for Cancer Research, Toronto, ON M5G 0A3, Canada; 22Department of Computer Science, University of Toronto, Toronto, ON M5S 2E4, Canada; 23The Centre for Applied Genomics, The Hospital for Sick Children, Toronto, ON M5G 0A4, Canada; 24Department of Statistical Sciences, University of Toronto, Toronto, ON M5G 1Z5, Canada; 25Division of Emergency Medicine, University of Toronto, Toronto, ON M5S 3H2, Canada

**Keywords:** SARS-CoV-2, COVID-19, hospitalization, risk factors, clinical characteristics, symptoms, comorbidities

## Abstract

The GENCOV study aims to identify patient factors which affect COVID-19 severity and outcomes. Here, we aimed to evaluate patient characteristics, acute symptoms and their persistence, and associations with hospitalization. Participants were recruited at hospital sites across the Greater Toronto Area in Ontario, Canada. Patient-reported demographics, medical history, and COVID-19 symptoms and complications were collected through an intake survey. Regression analyses were performed to identify associations with outcomes including hospitalization and COVID-19 symptoms. In total, 966 responses were obtained from 1106 eligible participants (87% response rate) between November 2020 and May 2022. Increasing continuous age (aOR: 1.05 [95%CI: 1.01–1.08]) and BMI (aOR: 1.17 [95%CI: 1.10–1.24]), non-White/European ethnicity (aOR: 2.72 [95%CI: 1.22–6.05]), hypertension (aOR: 2.78 [95%CI: 1.22–6.34]), and infection by viral variants (aOR: 5.43 [95%CI: 1.45–20.34]) were identified as risk factors for hospitalization. Several symptoms including shortness of breath and fever were found to be more common among inpatients and tended to persist for longer durations following acute illness. Sex, age, ethnicity, BMI, vaccination status, viral strain, and underlying health conditions were associated with developing and having persistent symptoms. By improving our understanding of risk factors for severe COVID-19, our findings may guide COVID-19 patient management strategies by enabling more efficient clinical decision making.

## 1. Introduction

Since the beginning of the coronavirus disease 2019 (COVID-19) pandemic, a diverse spectrum of clinical presentations has been identified among individuals infected by severe acute respiratory syndrome coronavirus 2 (SARS-CoV-2), ranging from asymptomatic infection to severe illness resulting in hospitalization or death [1]. Specific risk factors for COVID-19 severity have been described by numerous studies and include male sex, increasing age and body mass index (BMI), and underlying chronic health conditions [2,3,4,5,6,7,8]. Meta-analyses which have consolidated the findings of several studies have reported underlying health conditions such as hypertension, diabetes, respiratory disease, and cardiovascular disease to be associated with increased risk of severe illness or mortality [9,10]. However, there is considerable heterogeneity among risk factors identified between studies, which may be attributed in part to variability in study design and differences between populations. Moreover, many studies have only examined risk factors for severity within the context of hospitalized patients and have not compared these individuals to the majority who do not require treatment [2,3,4,7]. By characterizing individuals with milder illness, risk factors for severe illness leading to hospitalization can be identified more appropriately. COVID-19 symptom prevalence and persistence have also been characterized across many populations, and significant heterogeneity has been observed in both within- and between-country study populations [11,12,13]. Despite this heterogeneity, these independent studies have found that individuals with symptomatic illness most commonly report experiencing fever, cough, and fatigue. However, little work has been performed to assess whether patient factors contribute to the development or persistence of these symptoms. By identifying associations between patient factors and clinical manifestations of COVID-19, features of patients may be utilized to predict the progression of illness.

GENCOV is a prospective, observational cohort study of COVID-19-positive adults across the Greater Toronto Area in Ontario, Canada, which seeks to identify patient characteristics associated with differences in COVID-19 severity and patient outcomes. Here, our objective was to (1) identify patient characteristics associated with hospitalization from COVID-19 and (2) examine COVID-19 symptoms, their persistence, and associations with hospitalization and other patient factors. The limited generalizability of previous studies highlights the need for population-specific considerations when assessing COVID-19 patients and their clinical presentation and determining the likely course of their illness. By improving our understanding of risk factors for (1) hospitalization, (2) severe clinical presentations, and (3) persistent illness, clinical decision-making processes for COVID-19 patients can be refined to facilitate targeted and rapid patient care. Based on prior studies which have investigated risk factors for both symptomatic and more severe COVID-19 [2,3,14], we hypothesized that both intrinsic and extrinsic patient factors are associated with variable COVID-19 severity and that these same factors influence the presentation and persistence of specific clinical manifestations of acute COVID-19.

## 2. Materials and Methods

### 2.1. Participant Recruitment

Participants were recruited across the Greater Toronto Area in Ontario, Canada, at one of the following sites: Sinai Health System, University Health Network (including Toronto General Hospital and Toronto Western Hospital), William Osler Health System (including Brampton Civic Hospital and Etobicoke General Hospital), Mackenzie Health (including Cortellucci Vaughan Hospital and Mackenzie Richmond Hill Hospital), and Women’s College Hospital. Hospitalized inpatients who were admitted with or from COVID-19 illness were prospectively recruited into the study. Outpatients were recruited from the emergency department or COVID-19 assessment centres at participating hospital sites. Study participants with a prior SARS-CoV-2 infection were retrospectively recruited using study fliers posted at participating hospital sites. Hospitalization status was confirmed at enrolment. Participants were enrolled between November 2020 and May 2022 based on the following inclusion criteria: (1) 18 years or older, (2) had a confirmed positive SARS-CoV-2 PCR test, and (3) provided informed consent to participate in the study.

### 2.2. Patient Data Collection by Intake Surveys

Upon enrolment, participants were provided with a link to an intake survey [15] hosted on NOVI Survey’s web-based survey software (version 5.9.4510, 3rd Millennium Inc., Cambridge, MA, USA). The survey was developed and refined according to feedback provided by the GENCOV study team, consisting of 21 co-investigators and researchers across multiple healthcare disciplines. Participants completed the survey in English and were permitted to pause, resume, and modify responses prior to submission. Data were collected between 9 and 411 days (median: 86 days) post-COVID-19 diagnosis and stored on REDCap (version 11.4.2, Vanderbilt University, Nashville, TN, USA), a secure database hosted on Mount Sinai Hospital’s server. Patient-reported information related to demographics, medical history, and COVID-19 symptoms (type and duration) was collected. Information related to COVID-19 vaccination was also obtained from participants, including the total number of doses received, the date of each dose, and dose manufacturers. With respect to patient medical histories, data related to 14 categories of health conditions (shown in Table A1) were collected. Participants who reported having any of the listed conditions were prompted to elaborate on their specific conditions in free-text format. A manual review of free-text responses was performed to correct entry errors or misclassification. Clinical conditions which could be categorized in more than one way (e.g., Hepatitis B infection as both a non-SARS-CoV-2 viral infection and a hepatic condition) were subcategorized (e.g., viral hepatitis) and were included under all relevant conditions to ensure responses were categorized consistently (Table A1). Participants were presented with a list of 23 COVID-19 symptoms and were asked whether they experienced each symptom, and if so, for how long. Symptom durations were reported to the nearest 1-week interval (e.g., <1 week, 1–2 weeks), with a maximum reportable duration of >8 weeks. Data from 47 participants who completed the survey within 8 weeks of COVID-19 positivity and reported ongoing symptoms were excluded from symptom duration analyses.

### 2.3. Collection of Viral Lineage Data

SARS-CoV-2 viral genomes were isolated from nasopharyngeal swabs collected at baseline and subsequently sequenced as per the study protocol [15]. Viral lineages which were incompletely sequenced were inferred by spike (S)-gene target failure (SGTF) and/or epidemiological data of circulating variants during the study period [16]. Deletion of amino acids 69 and 70 within the S gene of SARS-CoV-2, sometimes attributable to the N501Y mutation, can result in SGTF for some real-time reverse transcriptase polymerase chain reaction (RT-PCR) testing methods. A subset of study samples were tested using the TaqPath COVID-19 PCR (Thermo Fisher Scientific, Waltham, MA, USA), which demonstrates SGTF for SARS-CoV-2 viral types Alpha (lineage B.1.1.7) and Omicron (lineages B.1.1.529, BA.1, BA.2, XBB). SGTF was defined as non-detection of the S-gene target among samples that tested positive (cycle threshold < 37) for both the N-gene and ORF1ab-gene targets. In this study, SGTF was used as an indicator of presumptive Alpha or Omicron viral type in the absence of sequence data. This was accomplished by reviewing the epidemiological data for circulating variants at the time of sample collection using a publicly available Nextstrain database maintained by Public Health Ontario and populated using data generated by the Ontario COVID-19 Genomics Network [17,18]. Based on these data, the Alpha viral type was in circulation within the population between December 2020 and August 2021, while Omicron was in circulation beginning November 2021, based on the sample collection date. As there is no overlap between the circulation of these two viral types, collection date in conjunction with SGTF is a strong predictor of Alpha and Omicron viral types for their respective periods of circulation.

Further discrimination of SARS-CoV-2 variants was accomplished for a subset of samples using a multiplex real-time RT-PCR assay. The assay discriminates between Alpha and Beta/Gamma viral types by detecting site-specific mutations in the SARS-CoV-2 spike gene by identifying three targets: (1) N501 (wild-type) due to the presence of adenine at nucleic acid position 23063, (2) N501Y (mutation associated with Alpha, Beta, and Gamma viral types) due to presence of A23063T substitution, and (3) E484K (mutation associated with Beta and Gamma viral types) due to presence of G23012A substitution. Reactions demonstrating N501Y mutation in the absence of E484K mutation were designated Alpha, while those demonstrating N501Y and E484K were designated Beta/Gamma (further discrimination was not possible without sequencing).

### 2.4. Statistical Analyses

Data analyses were performed using Microsoft Excel (version 2307, Microsoft Corporation, Redmond, WA, USA), R (version 4.1.1, The R Foundation for Statistical Computing, Vienna, Austria), and StataSE (version 14.0, StataCorp LLC, College Station, TX, USA). Pearson’s chi-squared test or Fisher’s exact test (for comparisons with fewer than 5 expected observations) was performed in R version 4.1.1 using the “chisq.test()” and “fisher.test()” functions to compare COVID-19 symptoms based on hospitalization status. The Mann–Whitney U test was performed in Stata 14 using the “ranksum” function to compare symptom durations between patient groups based on participants’ hospitalization status. Symptoms were excluded from duration analysis if they were reported by fewer than 10 total respondents (e.g., seizures) or less than 10% of either patient group after stratification (e.g., seizures, hemoptysis, conjunctivitis, ear pain, skin rash).

When examining patient characteristics as risk factors for hospitalization, the following were controlled for as covariates based on previous evidence of being associated with COVID-19 severity and mortality [8,9,10,19]: sex (male vs. female), age, BMI, ethnicity (White/European vs. non-White/European), and underlying health conditions including hypertension, diabetes, and cardiovascular and pulmonary conditions. The following characteristics were analyzed as potential risk factors for developing and having persistent COVID-19 symptoms based on prior evidence of being predictive of symptomatic illness [20,21,22,23,24,25,26]: sex (male vs. female), age, BMI, ethnicity (White/European vs. non-White/European), viral strain (wild-type vs. variant), vaccination status (unvaccinated vs. vaccinated), and having any underlying health conditions.

Logistic regression was performed to model the relationship of several covariates with dichotomous outcomes including hospitalization and COVID-19 symptom prevalence. Logistic regression was performed in Stata 14 using the “logistic” function to fit maximum-likelihood dichotomous logistic models. Crude (OR) and adjusted (aOR) odds ratios with 95% confidence intervals were estimated for univariable and multivariable analyses, respectively.

Interval regression analysis (a type of censored regression) was used to model the relationship between risk factors for symptomatic COVID-19 and the persistence of symptoms. Interval regression was performed in Stata 14 using the “intreg” function to fit linear models with an outcome measured as point data, interval data, left-censored data, or right-censored data. The duration of each symptom was analyzed as a separate outcome. The durations for seizures, conjunctivitis, and hemoptysis were excluded from multivariable regression analysis due to insufficient observations for each symptom. Regression coefficients (β) and 95% confidence intervals correspond to the estimated difference in symptom duration in weeks. For all multivariable regression analyses, only observations with complete data for all covariates were included in the final models. The statistical significance level was set at α = 0.05.

## 3. Results

### 3.1. Participant Characteristics and Associations with Hospitalization

In total, 966 responses were obtained from 1106 eligible participants (87% response rate; Table 1). Most respondents were outpatients (94.4%), female (56.8%), of White or European ethnicity (50.1%), had at least one underlying medical condition (57.1%), and had no history of smoking (60.2%). The most prevalent non-White/European ethnicities represented among respondents included Middle Eastern (9.8%) and South Asian (7.8%) (Table A2). The median age of respondents was 43 years (IQR: 32–55 years), while the median body mass index (BMI) was determined to be 25.9 kg/m^2^ (IQR: 22.9–29.1 kg/m^2^). Most respondents indicated that they were unvaccinated prior to having COVID-19 (69.1%), while only 25.3% of respondents were vaccinated. A summary of the number of vaccine doses, time of most recent vaccination, and specific vaccine combinations received by participants is presented in Table A3. The remainder of the participants (5.6%) did not disclose their vaccination status at enrolment. Approximately 28.7% of participants were determined to have been infected by wild-type SARS-CoV-2. The most prevalent viral variants detected among respondents were Alpha (20.8%), Omicron (7.1%), and Delta (6.6%) (Table A2). Viral lineages were undetermined for 336 (34.8%) participants due to the unavailability of viral swabs or complete sequencing data.

Older age, increasing BMI, non-White/European ethnicity, and infection by SARS-CoV-2 variants were associated with increased odds of hospitalization by univariable and multivariable analysis (controlling for covariates described above). Univariable analysis, but not multivariable analysis, identified vaccination prior to infection was associated with reduced odds of hospitalization (Table 1). Vaccination status was not included in the multivariable analysis because all vaccinated participants with complete data were outpatients. Similarly, having any underlying health conditions was excluded from multivariable analysis given the collinearity with other predictors. Sex, having any non-specific underlying health conditions, and history of smoking were not associated with hospitalization risk.

The most prevalent health conditions among respondents included gastrointestinal disorders (14.7%), hypertension (12.7%), and endocrine disorders (12.1%) including diabetes (6.3%) (Table 2). Univariable analysis identified that hypertension, endocrine conditions including diabetes, cardiovascular conditions, cancer, lipid conditions, and hepatic conditions were significantly associated with increased risk of hospitalization. After controlling for covariates, hypertension was the only condition which remained significantly associated with hospitalization. Blood, gastrointestinal, pulmonary, autoimmune, neurologic/psychiatric, renal, and non-SARS-CoV-2 viral infections were not associated with hospitalization. Hereditary genetic conditions were not assessed as a risk factor for hospitalization due to their low prevalence among the study cohort.

### 3.2. COVID-19 Symptom Prevalence, Duration, and Associations with Patient Characteristics

With respect to the 23 COVID-19 symptoms documented (Table 3), fatigue (80.0%), headache (64.8%), and muscle aches (63.7%) were the most frequently reported symptoms, while conjunctivitis (2.9%), cough with bloody sputum/phlegm (i.e., hemoptysis) (2.2%), and seizures (0.4%) were the most uncommon symptoms. The longest persisting symptoms included shortness of breath, loss of taste, and loss of smell, which lasted 3–4 weeks on average (Figure 1). Short-lived symptoms included fever, sore throat, diarrhea, and conjunctivitis, lasting <1 week on average. When stratified by hospitalization status, shortness of breath, fever, runny nose/nasal congestion, chest pain, wheezing, altered consciousness/confusion, abdominal pain, and vomiting/nausea were significantly associated with hospitalization (Table 3). Conversely, loss of smell was significantly more prevalent among outpatients. Furthermore, inpatients reported significantly longer-lasting symptoms compared to outpatients for fever, cough (productive and non-productive), sore throat, runny nose/nasal congestion, chest pain, muscle aches, joint pain, fatigue, shortness of breath, headache, and abdominal pain (Figure 2).

Sex, BMI, age, ethnicity, viral strain, vaccination status, and having one or more underlying health conditions were independently associated with the development of one or more COVID-19 symptoms (Figure 3). Female sex was independently associated with greater odds of developing sore throat, runny nose/nasal congestion, wheezing, chest pain, headache, abdominal pain, and vomiting/nausea. Increasing BMI was independently associated with greater odds of developing productive and non-productive cough, wheezing, chest pain, shortness of breath, and diarrhea. Older age was independently associated with lower odds of developing productive and non-productive cough, sore throat, runny nose/nasal congestion, chest pain, headache, and loss of taste and smell. Non-White or non-European ethnicity was associated with greater odds of developing sore throat and conjunctivitis. Infection by SARS-CoV-2 viral variants was associated with lower odds of loss of taste and smell, and greater odds of fever, productive and non-productive cough, chest pain, vomiting/nausea, and skin rash. Vaccination prior to infection was associated with lower odds of developing productive cough, chest pain, shortness of breath, and diarrhea. However, vaccination was also associated with greater odds of experiencing runny nose/nasal congestion. Lastly, having one or more underlying health conditions was independently associated with greater odds of productive and non-productive cough, runny nose/nasal congestion, chest pain, fatigue, headache, altered consciousness/confusion, abdominal pain, and diarrhea. None of the covariates examined were associated with developing hemoptysis, ear pain, muscle aches, joint pain, or seizures.

Sex, BMI, age, ethnicity, viral strain, vaccination status, and having one or more underlying health conditions were similarly associated with differences in the duration of COVID-19 symptoms (Figure 4). Female sex was predictive of longer-lasting symptoms including runny nose/nasal congestion, chest pain, and muscle aches. Increasing BMI was also predictive of longer-lasting fever, muscle aches, joint pain, and fatigue, while older age was predictive of longer-lasting fever, productive and non-productive cough, fatigue, and abdominal pain. Non-White or non-European ethnicity was predictive of longer-lasting muscle aches, headache, and diarrhea. Infection by viral variants was predictive of shorter-lasting altered consciousness or confusion. Vaccination prior to infection was predictive of shorter-lasting fever and muscle aches and longer-lasting diarrhea. Lastly, having one or more comorbidities was associated with longer-lasting chest pain, muscle aches, joint pain, fatigue, and shortness of breath. None of the covariates were associated with differences in the duration of sore throat, ear pain, wheezing, vomiting/nausea, skin rash, or loss of taste or smell.

## 4. Discussion

Health service interruptions resulting from the COVID-19 pandemic have brought attention to the growing need for better strategies to prioritize and allocate resources effectively during periods of increased strain on healthcare systems. In the context of COVID-19, we propose that (1) identifying patient features which are predictive of illness requiring hospitalization and (2) characterizing clinical symptoms and complications associated with severe illness are critical to managing and improving patient care. Our study provides unique insights into improving the clinical decision-making process for COVID-19 patients by examining these measures collectively in a cohort of patients possessing diverse health histories, clinical characteristics, and SARS-CoV-2 viral variants.

We have identified several risk factors independently associated with hospitalization which corroborate the findings of previous studies. Specifically, patient factors including non-White/European ethnicity, increasing age, and higher BMI have similarly been shown to be associated with more severe illness and hospitalization. A systematic review and meta-analysis conducted by Sze et al. [27] found that ethnic minority groups (specifically individuals from Black and Asian communities) were at greater risk of COVID-19 infection compared to White individuals. Consequently, they also identified that Asian individuals were at greater risk of mortality, owing to increased transmission among members of this community due to various lifestyle and social factors. Similarly, Magesh et al. [28] found that members of racial and ethnic minority groups across 68 independent studies were at heightened risk of COVID-19 positivity and disease severity. In a review article published by Gao et al. [8], older age and BMI (as it relates to obesity) were highlighted as prominent risk factors for disease severity across a series of multivariable-adjusted analyses. Studies comparing illness severity resulting from infection by SARS-CoV-2 variants of concern have found variants including Alpha (B.1.1.7) and Delta (B.1.617.2) to be associated with a greater risk of poorer outcomes relative to the wild-type virus [29,30]. As the Alpha variant was the most highly represented SARS-CoV-2 variant in our cohort, the significant association observed with respect to hospitalization is congruent with other studies. Similar to our study, studies conducted by Buchan et al. [31] and Moghadas et al. [32] have shown that vaccination drastically reduces the risk of adverse COVID-19 outcomes including hospitalization, even in cases of infection by SARS-CoV-2 variants such as Delta and Omicron.

In contrast to other studies [9,10], associations identified between specific underlying health conditions (e.g., endocrine conditions including diabetes, cancer) and hospitalization were no longer significant after controlling for covariates, with the exception of hypertension. Our findings suggest that other patient characteristics, such as age and BMI, are the greatest risk factors for hospitalization among participants of our study. Moreover, unlike other studies [33], we did not observe any significant difference in the risk of hospitalization between sexes. The incongruence of these findings highlights how certain risk factors for COVID-19 severity may be population-specific and dependent on study design. For example, differences in health behaviours between sexes are suggested to play a role in the severity and outcomes of COVID-19. Handwashing, masking, and adherence to public health guidance are more likely to be followed by females, while males have been reported to be more likely to delay accessing healthcare resources [34,35]. Some studies have also hypothesized that social factors including gender-linked occupations and structural exposures (e.g., incarceration and homelessness) may contribute to sex disparities in COVID-19 outcomes [35]. As a result, whether these social determinants are present is likely to be population-specific and may contribute to differences in risk factors identified between study populations. Similarly, health conditions which are more prevalent among individuals of a certain population may play a greater role in patient outcomes and consequently may be identified as significant risk factors for COVID-19.

In addition to patient characteristics, COVID-19 symptoms are also suggested to have prognostic value [36]. Symptoms including shortness of breath and fever were shown to be associated with hospitalization, and they have similarly been found to be common clinical presentations among those with severe illness by other studies [12,37]. Conversely, loss of smell was identified as the only symptom more prevalent among outpatients of our study and is suggested to be an early predictor of mild COVID-19 given its potential link to rapid antiviral responses within the nasal epithelium [38].

The associations identified between participant characteristics and specific symptoms provide novel insights into risk factors for developing or having persistent COVID-19 symptoms. Female sex was associated with greater odds of developing many of the symptoms examined, which is supported by previous studies. For example, Lhendup et al. found that males were 64% less likely to be symptomatic compared to females [20]. Similarly, our findings provide supporting evidence that individuals with comorbidities, individuals with higher BMI (specifically those who are clinically overweight or obese), and those who are non-White or non-European are more likely to develop symptomatic illness and experience a greater number of symptoms over the course of their illness. Yu et al. [26] found that the presence of comorbidities such as hypertension was predictive of symptomatic progression among patients who were asymptomatic at admission. Similarly, Cheng et al. [22] found that individuals classified as overweight or obese tended to experience a greater number of severe symptoms including shortness of breath relative to those who were not classified as overweight or obese. In a study conducted by Patel and colleagues [39], non-Hispanic Black and Hispanic participants were found to be more frequently affected by symptoms compared to non-Hispanic White participants. The associations we identified between the development of specific symptoms and SARS-CoV-2 viral variants are consistent with differences in the clinical presentation of patients infected by SARS-CoV-2 viral variants characterized previously [24]. We also found that COVID-19 vaccination prior to infection was associated with lower odds of developing more severe symptoms (e.g., shortness of breath) and greater odds of developing mild symptoms (e.g., nasal congestion), which provides evidence in support of the protection conferred by vaccines against severe symptomatic infection [25].

As we found age to be a risk factor for hospitalization, the independent associations observed between increasing age and lower odds of developing specific COVID-19 symptoms were unexpected. Indeed, previous studies have reported that older individuals are at greater risk of developing symptomatic COVID-19 [21]. However, as the older individuals examined in this study only included those who survived their illness (most of whom were outpatients and never hospitalized), our observations are consistent with favourable illness outcomes. Our findings suggest that there may be other factors (e.g., serological or genetic differences) which we have not assessed among older participants that aid in protecting against developing specific symptoms relative to younger adults. Additionally, as we assessed age as a continuous variable, the association identified between increasing age and lower symptom prevalence is relevant to the entire age range of all participants and not necessarily limited to elderly patients.

With respect to COVID-19 symptom duration, studies of long COVID-19 have similarly found that the most persistent symptoms include fatigue, shortness of breath, and loss of smell [40,41]. These studies have also identified that female sex, higher BMI, belonging to an ethnic minority, and underlying comorbidities are risk factors for long COVID-19, while variability has been observed with respect to age. In totality, our findings elucidate and provide evidence of risk factors for symptomatic and persistent illness. The significant associations we have identified suggest that both patient characteristics and clinical presentation should be utilized to inform decisions for patient care.

As most study variables were patient-reported measures, our study has a few limitations. Since the survey was conducted online and only in English, many of the collected variables of interest were dependent on participants’ ability and willingness to disclose all information to the study team (i.e., ability to access, comprehend, and complete the survey). Consequently, our data collection methods may have excluded or deterred non-English-speaking persons or non-native English speakers from participating in the survey. Moreover, as many of our outcomes of interest were patient-reported and obtained retrospectively, we recognize that this may be a potential source of bias in our findings. Data which may be affected by recall bias may include outcomes related to medical histories and COVID-19 symptoms. As participant recruitment spanned approximately a year and a half, the availability of COVID-19 vaccines and therapeutics improved over the course of the study period [42]. Conversely, the general accessibility of healthcare resources may have fluctuated depending on the level of strain on healthcare systems throughout various waves of the pandemic. Each of these factors may have contributed to differences in patients’ COVID-19 illness progression and outcomes. However, the impact of these factors is highly context- and patient-specific (e.g., the capacity of any given hospital, or the use of COVID-19 therapeutics), and as a result, we were unable to thoroughly assess their effect on our outcomes of interest. Similarly, we did not assess other non-physiological patient factors such as COVID-19 vaccine hesitancy attitudes and public health misinformation which may be related to COVID-19 severity and outcomes [43,44,45]. However, these factors were unlikely to have affected or biased recruitment, as most inpatients and outpatients were enrolled within the same period between January 2021 and July 2021 (Figure A1). Due to the limited availability of data for deceased participants enrolled in the study, these individuals were not included in our analyses. Consequently, further stratification by clinical course (e.g., ICU admission) or outcomes (e.g., death) was not performed. Additionally, due to the complexity of COVID-19 vaccination among participants of our study, we were limited in our ability to examine associations between different vaccine combinations and the outcomes of interest. As different vaccine regimens and more recent vaccination have been shown to impact symptomatic COVID-19 [46], additional differences may exist in the clinical characteristics between vaccinated participants which we were unable to account for.

## 5. Conclusions

In summary, in a diverse cohort of COVID-19-positive adults residing in Ontario, Canada, we have identified (1) risk factors associated with hospitalization and (2) predictors of COVID-19 symptom development and persistence. Specifically, we have shown that intrinsic patient factors including age, ethnicity, BMI, and comorbidities such as hypertension and extrinsic factors including viral strain are risk factors for hospitalization. These findings are congruent with those of previous studies and lend support to the body of evidence surrounding risk factors for COVID-19 severity. More importantly, however, we have shown that these same patient factors, in addition to other factors including sex and vaccination status, are associated with the differences in the prevalence and persistence of specific clinical manifestations of COVID-19. These findings may be utilized to provide insight into the likely course of one’s illness based on patient characteristics. Furthermore, the findings of this study may aid in improving COVID-19 patient management strategies which will enable more efficient clinical decision making.

## Figures and Tables

**Figure 1 viruses-15-01764-f001:**
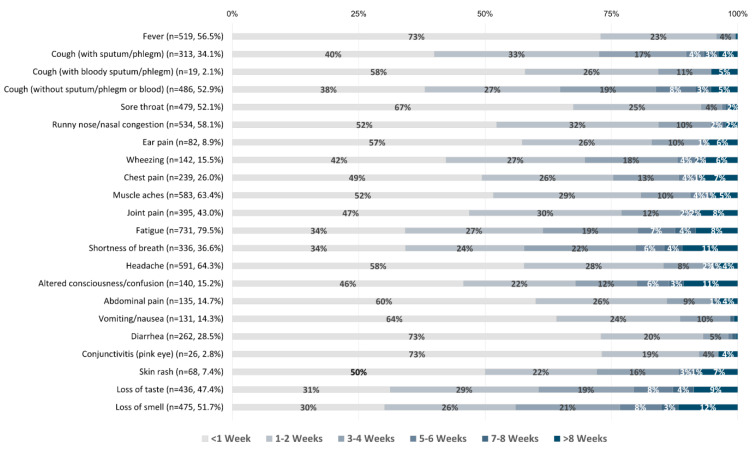
COVID-19 symptom durations reported by all respondents (in weeks). n values and percentages indicated next to each symptom reflect the number and proportion of participants who experienced a given symptom out of the entire cohort. Percentages indicated on bars represent the proportion of symptomatic participants who reported a given symptom duration.

**Figure 2 viruses-15-01764-f002:**
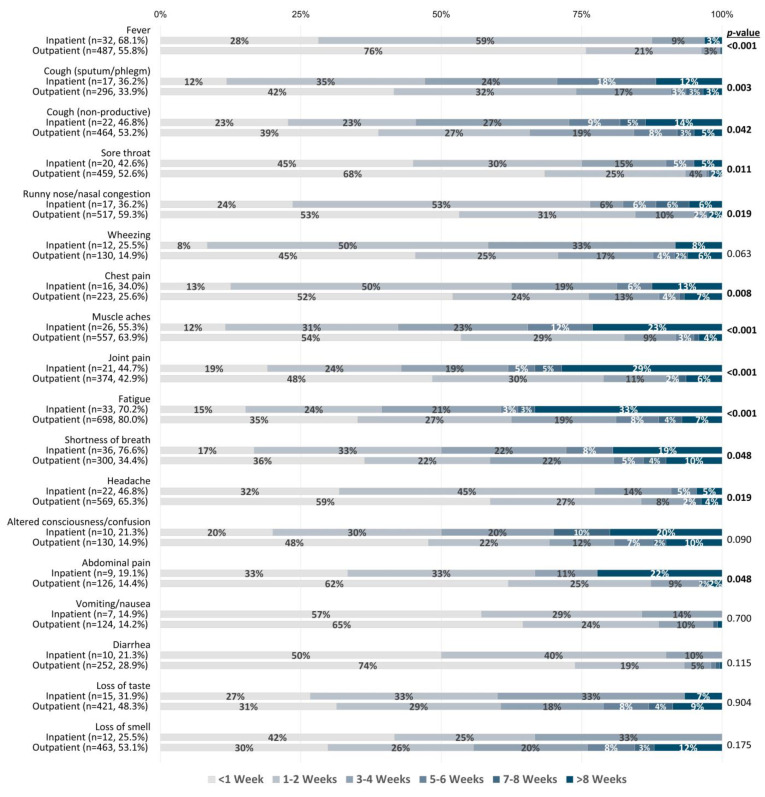
COVID-19 symptom duration (in weeks) stratified by hospitalization status. Comparisons of symptom duration between patient groups were performed using the Mann–Whitney U test. Symptoms reported by >10% of each patient group were included for analysis. n values and percentages indicated below each symptom reflect the total number and proportion of participants out of each patient group who experienced a given symptom. Percentages indicated on bars represent the proportion of symptomatic participants from each patient group who reported a given symptom duration.

**Figure 3 viruses-15-01764-f003:**
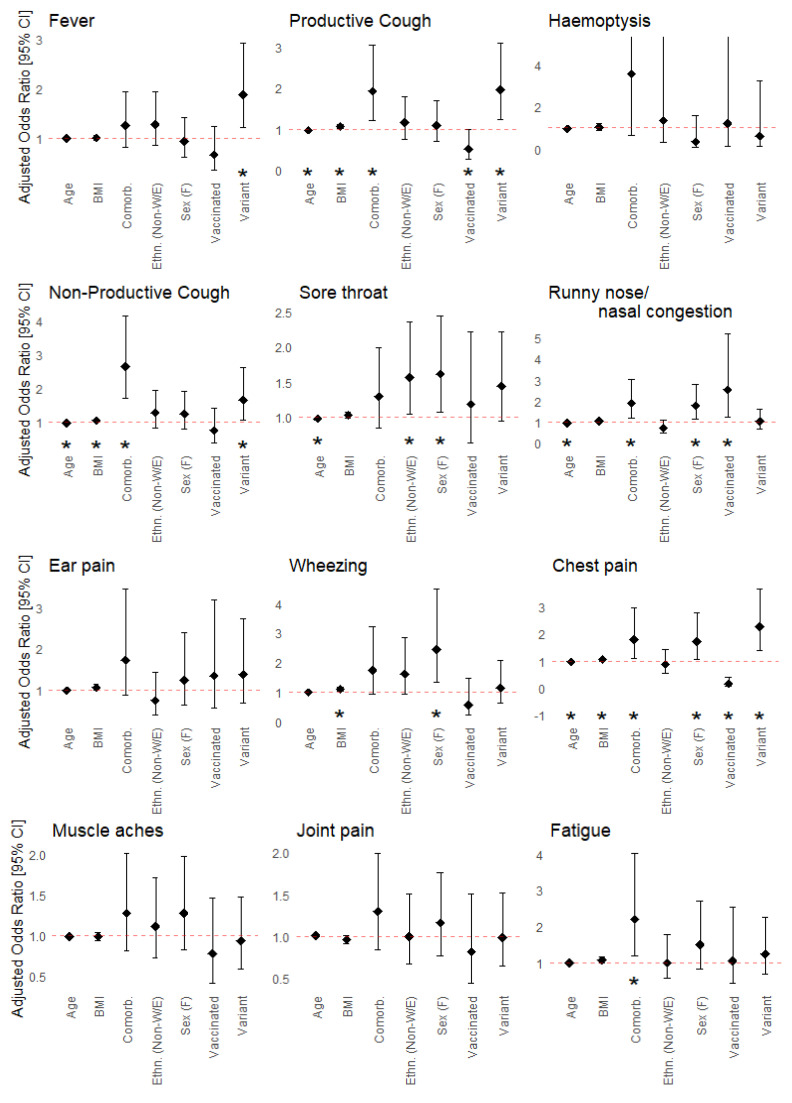
Associations between symptom prevalence and patient characteristics including sex (male (reference) vs. female), age, BMI, ethnicity (White/European (reference) vs. Non-White/European), viral variant (wild-type (reference) vs. viral variant), vaccination status (unvaccinated (reference) vs. vaccinated), and underlying medical conditions (none (reference) vs. any). Adjusted odds ratios and 95% confidence intervals for each covariate were calculated using a multiple logistic regression model. Adjusted odds ratios correspond to the odds of experiencing a given symptom in the comparator group relative to the reference group (for categorical variables), or for a one-year increase in age or one-unit increase in BMI. Statistically significant associations are indicated by asterisks (*).

**Figure 4 viruses-15-01764-f004:**
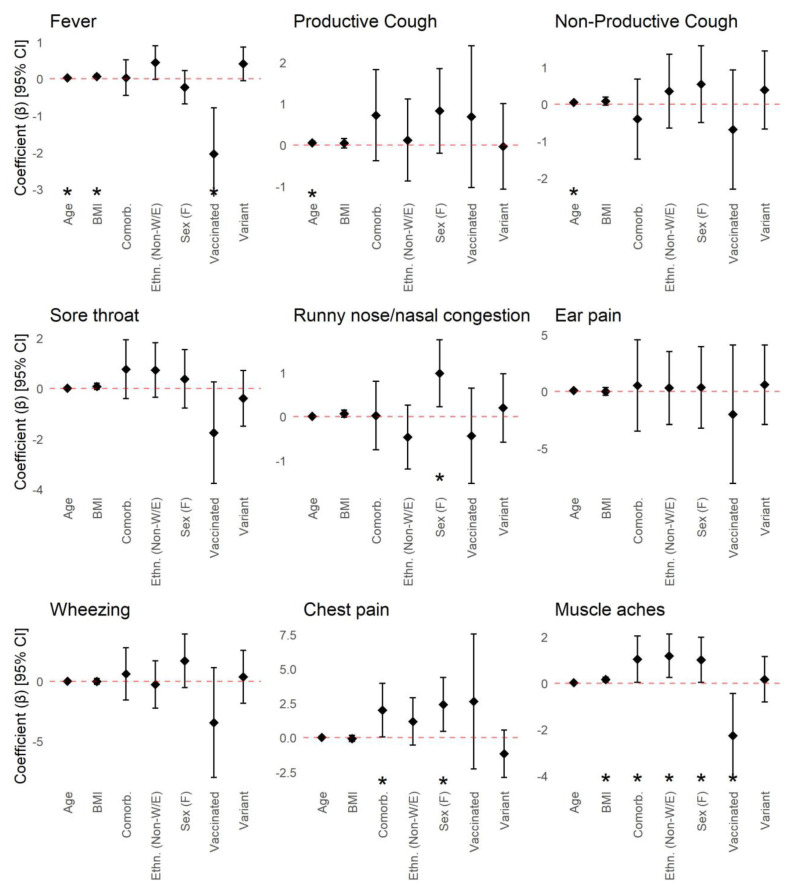
Associations between symptom duration and patient characteristics including sex (males (reference) vs. females), age, BMI, ethnicity (White/European (reference) vs. Non-White/European), viral variant (wild-type (reference) vs. viral variant), vaccination status (unvaccinated (reference) vs. vaccinated), and underlying medical conditions (none (reference) vs. any). Regression coefficients and 95% confidence intervals were calculated using an interval regression model. Regression coefficients correspond to the difference in symptom duration (in weeks) compared to the reference group (for categorical variables), or for a one-year increase in age or one-unit increase in BMI. Statistically significant associations are indicated by asterisks (*).

**Table 1 viruses-15-01764-t001:** Summary of GENCOV participant characteristics, stratified by hospitalization status. *p*-values, crude odds ratios (ORs), adjusted odds ratios (aORs), and 95% confidence intervals were calculated from the logistic regression models. Variables which were controlled in the multiple logistic regression model included continuous age and BMI, sex, ethnicity (White/European vs. non-White/European), hypertension, diabetes, cardiovascular conditions, and pulmonary conditions. Percentages may not add up to 100% due to rounding.

	No. of Observations (%)						
				Univariable		Multivariable	
Characteristics	Total (*n* = 966)	Inpatient (*n* = 54)	Outpatient (*n* = 912)	*p* (OR)	OR [95%CI]	*p* (aOR)	aOR [95%CI]
**Sex**							
Male	417 (43.2)	28 (51.8)	389 (42.6)	0.187	0.69 [0.40–1.20]	0.977	1.01 [0.47–2.18]
Female	549 (56.8)	26 (48.2)	523 (57.4)				
**Age (Median [IQR])**	43 [32–55]	56 [46–65]	42 [31–54]	**<0.001**	**1.07** **[1.05–1.09]**	**0.007**	**1.05** **[1.01–1.08]**
<65	882 (91.3)	38 (70.4)	844 (92.5)				
≥65	84 (8.7)	16 (29.6)	68 (7.5)				
**Ethnicity**							
White/European	484 (50.1)	15 (27.8)	469 (51.4)	**0.002**	**2.75** **[1.47–5.14]**	**0.014**	**2.72** **[1.22–6.05]**
Non-White/European	408 (42.2)	33 (61.1)	375 (41.1)				
Mixed Ethnicity	48 (5.0)	1 (1.8)	47 (5.2)				
Unknown	26 (2.7)	5 (9.3)	21 (2.3)				
**BMI (Median [IQR])**	25.9 [22.9–29.1]	30.2 [26.6–34.6]	25.8 [22.8–28.7]	**<0.001**	**1.18** **[1.12–1.24]**	**<0.001**	**1.17** **[1.10–1.24]**
<25 kg/m^2^	303 (31.4)	4 (7.4)	299 (32.8)				
≥25 kg/m^2^	419 (43.4)	35 (64.8)	384 (42.1)				
Unknown	244 (25.2)	15 (27.8)	229 (25.1)				
**Viral Variants**							
Wild-Type	277 (28.7)	7 (13.0)	270 (29.6)	**0.045**	**2.44** **[1.02–5.83]**	**0.012**	**5.43** **[1.45–20.34]**
Variant	353 (36.5)	21 (38.9)	332 (36.4)				
Unknown	336 (34.8)	26 (48.1)	310 (34.0)				
**Vaccination Status** **(at COVID-19 positivity date)**							
Unvaccinated	668 (69.1)	44 (81.5)	624 (68.4)	**0.003**	**0.12** **[0.03–0.49]**	N/A	N/A
Vaccinated (Any dose)	244 (25.3)	2 (3.7)	242 (26.5)				
Unknown	54 (5.6)	8 (14.8)	46 (5.1)				
**Underlying Health Conditions**							
None	414 (42.9)	18 (33.3)	396 (43.4)	0.148	1.53 [0.86–2.74]	N/A	N/A
Any (≥1)	552 (57.1)	36 (66.7)	516 (56.6)				
**History of Smoking**							
Yes	357 (37.0)	16 (29.6)	341 (37.4)	0.565	0.83 [0.45–1.55]	0.745	1.14 [0.52–2.52]
No	582 (60.2)	31 (57.4)	551 (60.4)				
Unknown	27 (2.8)	7 (13.0)	20 (2.2)				

**Table 2 viruses-15-01764-t002:** Summary of GENCOV participant self-reported health conditions, stratified by hospitalization status. *p*-values, crude odds ratios (ORs), adjusted odds ratios (aORs), and 95% confidence intervals were calculated from the logistic regression models. Variables which were controlled in the multiple logistic regression model included continuous age and BMI, sex, ethnicity (White/European vs. non-White/European), hypertension, diabetes, cardiovascular conditions, and pulmonary conditions. Percentages may not add up to 100% due to rounding.

	No. of Observations (%)						
				Univariable		Multivariable	
Health Condition	Inpatient (*n* = 54)	Outpatient (*n* = 912)	Total (*n* = 966)	*p* (OR)	OR [95%CI]	*p* (aOR)	aOR [95%CI]
**Hypertension**							
Yes	22 (40.7)	101 (11.1)	123 (12.7)	**<0.001**	**6.56** **[3.58–12.01]**	**0.015**	**2.78** **[1.22–6.34]**
No	26 (48.1)	783 (85.9)	809 (83.7)				
Unknown	6 (11.1)	28 (3.1)	34 (3.5)				
**Endocrine Conditions**							
Yes	17 (31.5)	100 (11.0)	117 (12.1)	**<0.001**	**4.20** **[2.25–7.83]**	0.343	1.91 [0.50–7.32]
Diabetes (Any)	13 (24.1)	48 (5.3)	61 (6.3)	**<0.001**	**6.37** **[3.17–12.80]**	0.677	0.79 [0.26–2.41]
Type 2 Diabetes	13 (24.1)	45 (4.9)	58 (6.0)				
Type 1 Diabetes	1 (1.9)	5 (0.5)	6 (0.6)				
Endocrine Cancer	1 (1.9)	5 (0.5)	6 (0.6)				
Endocrine Autoimmune Disorders	0 (0.0)	10 (1.1)	10 (1.0)				
Gestational Diabetes	0 (0.0)	2 (0.2)	2 (0.2)				
Maturity-Onset Diabetes of the Young	0 (0.0)	2 (0.2)	2 (0.2)				
Pre-Diabetes	0 (0.0)	2 (0.2)	2 (0.2)				
No	32 (59.3)	790 (86.6)	822 (85.1)				
Unknown	5 (9.3)	22 (2.4)	27 (2.8)				
**Cardiovascular Conditions**							
Yes	11 (20.4)	72 (7.9)	83 (8.6)	**<0.001**	**3.55** **[1.73–7.29]**	0.787	1.16 [0.39–3.44]
No	35 (64.8)	814 (89.3)	849 (87.9)				
Unknown	8 (14.8)	26 (2.9)	34 (3.5)				
**Cancer**							
Yes	9 (16.7)	47 (5.2)	56 (5.8)	**<0.001**	**4.14** **[1.89–9.05]**	0.523	1.46 [0.46–4.66]
No	39 (72.2)	843 (92.4)	882 (91.3)				
Unknown	6 (11.1)	22 (2.4)	28 (2.9)				
**Lipid Conditions**							
Yes	7 (13.0)	43 (4.7)	50 (5.2)	**0.006**	**3.32** **[1.41–7.82]**	0.503	1.49 [0.47–4.74]
No	41 (75.9)	835 (91.6)	876 (90.7)				
Unknown	6 (11.1)	34 (3.7)	40 (4.1)				
**Hepatic Conditions**							
Yes	6 (11.1)	35 (3.8)	41 (4.2)	**0.009**	**3.41** **[1.36–8.54]**	0.259	2.03 [0.59–6.93]
Viral Hepatitis	1 (1.9)	8 (0.9)	9 (0.9)				
No	43 (79.6)	855 (93.8)	898 (93.0)				
Unknown	5 (9.3)	22 (2.4)	27 (2.8)				
**Blood Conditions**							
Yes	5 (9.3)	91 (10.0)	96 (9.9)	0.927	1.05 [0.40–2.71]	0.471	0.53 [0.09–2.97]
Iron Disorders	4 (7.4)	69 (7.6)	73 (7.6)				
Blood Cancer	0 (0.0)	6 (0.7)	6 (0.6)				
Blood Autoimmune Disorders	0 (0.0)	3 (0.3)	3 (0.3)				
No	42 (77.8)	799 (87.6)	841 (87.1)				
Unknown	7 (13.0)	22 (2.4)	29 (3.0)				
**Gastrointestinal Conditions**							
Yes	5 (9.3)	137 (15.0)	142 (14.7)	0.326	0.62 [0.24–1.60]	0.837	0.89 [0.31–2.59]
Gastrointestinal Autoimmune Disorders	1 (1.9)	17 (1.9)	18 (1.9)				
Gastrointestinal Cancer	1 (1.9)	9 (1.0)	10 (1.0)				
No	44 (81.5)	752 (82.5)	796 (82.4)				
Unknown	5 (9.3)	23 (2.5)	28 (2.9)				
**Pulmonary Conditions**							
Yes	4 (7.4)	91 (10.0)	95 (9.8)	0.762	0.85 [0.30–2.43]	0.883	1.10 [0.32–3.81]
No	41 (75.9)	793 (87.0)	834 (86.3)				
Unknown	9 (16.7)	28 (3.1)	37 (3.8)				
**Autoimmune Conditions**							
Yes	4 (7.4)	60 (6.6)	64 (6.6)	0.706	1.23 [0.43–3.52]	0.264	2.15 [0.56–8.23]
No	45 (83.3)	827 (90.7)	872 (90.3)				
Unknown	5 (9.3)	25 (2.7)	30 (3.1)				
**Neurologic/Psychiatric Conditions**							
Yes	3 (5.6)	64 (7.0)	67 (6.9)	0.853	0.89 [0.27–2.96]	0.834	1.18 [0.25–5.66]
Neurological Autoimmune Disorders	1 (1.9)	0 (0.0)	1 (0.1)				
No	43 (79.6)	819 (89.8)	862 (89.2)				
Unknown	8 (14.8)	29 (3.2)	37 (3.8)				
**Renal Conditions**							
Yes	3 (5.6)	19 (2.1)	22 (2.3)	0.085	3.00 [0.86–10.52]	N/A	N/A
Renal Cancer	1 (1.9)	0 (0.0)	1 (0.1)				
Renal Autoimmune Disorders	0 (0.0)	2 (0.2)	2 (0.2)				
No	46 (85.2)	875 (95.9)	921 (95.3)				
Unknown	5 (9.3)	18 (2.0)	23 (2.4)				
**Non-SARS-CoV-2 Viral Infections**							
Yes	1 (1.9)	33 (3.6)	34 (3.5)	0.559	0.55 [0.07–4.10]	0.697	0.64 [0.07–6.04]
No	47 (87.0)	852 (93.4)	899 (93.1)				
Unknown	6 (11.1)	27 (3.0)	33 (3.4)				
**Hereditary Genetic Conditions**							
Yes	0 (0.0)	14 (1.5)	14 (1.4)	N/A	N/A	N/A	N/A
Hereditary Blood Disorders	0 (0.0)	8 (0.9)	8 (0.8)				
Hereditary Lipid Disorders	0 (0.0)	1 (0.1)	1 (0.1)				
No	50 (92.6)	879 (96.4)	929 (96.2)				
Unknown	4 (7.4)	19 (2.1)	23 (2.4)				
**Other Conditions**							
Yes	13 (24.1)	91 (10.0)	104 (10.8)				
Sleep Disorders	5 (9.3)	34 (3.7)	39 (4.0)				
Muscle/Joint Conditions	4 (7.4)	21 (2.3)	25 (2.6)				
Gynecological/Urological Conditions	4 (7.4)	14 (1.5)	18 (1.9)				
Dermatological Conditions	1 (1.9)	12 (1.3)	13 (1.3)				
Bone Conditions	0 (0.0)	9 (1.0)	9 (0.9)				
Ophthalmologic Conditions	0 (0.0)	8 (0.9)	8 (0.8)				
No	41 (75.9)	821 (90.0)	862 (89.2)				

**Table 3 viruses-15-01764-t003:** Summary of GENCOV participant self-reported COVID-19 symptoms and associations with hospitalization status. *p*-values were calculated using Pearson’s chi-squared test or Fisher’s exact test (for comparisons with fewer than 5 expected observations). Percentages may not add up to 100% due to rounding.

COVID-19 Symptoms	Inpatient (*n* = 54)	Outpatient (*n* = 912)	Total (*n* = 966)	*p*
**Shortness of breath**				
Yes	42 (77.8%)	328 (36.0%)	370 (38.3%)	**<0.001**
No	2 (3.7%)	510 (55.9%)	512 (53.0%)	
Unknown	10 (18.5%)	74 (8.1%)	84 (8.7%)	
**Fever**				
Yes	39 (72.2%)	512 (56.1%)	551 (57.0%)	**<0.001**
No	4 (7.4%)	363 (39.8%)	367 (38.0%)	
Unknown	11 (20.4%)	37 (4.1%)	48 (5.0%)	
**Fatigue**				
Yes	39 (72.2%)	734 (80.5%)	773 (80.0%)	0.135
No	3 (5.6%)	136 (14.9%)	139 (14.4%)	
Unknown	12 (22.2%)	42 (4.6%)	54 (5.6%)	
**Muscle aches**				
Yes	30 (55.6%)	585 (64.1%)	615 (63.7%)	0.644
No	12 (22.2%)	275 (30.2%)	287 (29.7%)	
Unknown	12 (22.2%)	52 (5.7%)	64 (6.6%)	
**Non-productive cough**				
Yes	28 (51.9%)	493 (54.1%)	521 (53.9%)	0.132
No	12 (22.2%)	357 (39.1%)	369 (38.2%)	
Unknown	14 (25.9%)	62 (6.8%)	76 (7.9%)	
**Headache**				
Yes	28 (51.9%)	598 (65.6%)	626 (64.8%)	0.644
No	10 (18.5%)	254 (27.9%)	264 (27.3%)	
Unknown	16 (29.6%)	60 (6.6%)	76 (7.9%)	
**Joint pain**				
Yes	23 (42.6%)	397 (43.5%)	420 (43.5%)	0.208
No	17 (31.5%)	442 (48.5%)	459 (47.5%)	
Unknown	14 (25.9%)	73 (8.0%)	87 (9.0%)	
**Sore throat**				
Yes	22 (40.7%)	483 (53.0%)	505 (52.3%)	0.702
No	19 (35.2%)	369 (40.5%)	388 (40.2%)	
Unknown	13 (24.1%)	60 (6.6%)	73 (7.6%)	
**Productive cough**				
Yes	20 (37.0%)	315 (34.5%)	335 (34.7%)	0.207
No	22 (40.7%)	516 (56.6%)	538 (55.7%)	
Unknown	12 (22.2%)	81 (8.9%)	93 (9.6%)	
**Runny nose/nasal congestion**				
Yes	20 (37.0%)	546 (59.9%)	566 (58.6%)	**0.033**
No	22 (40.7%)	309 (33.9%)	331 (34.3%)	
Unknown	12 (22.2%)	57 (6.3%)	69 (7.1%)	
**Chest pain**				
Yes	20 (37.0%)	239 (26.2%)	259 (26.8%)	**0.004**
No	20 (37.0%)	590 (64.7%)	610 (63.1%)	
Unknown	14 (25.9%)	83 (9.1%)	97 (10.0%)	
**Wheezing**				
Yes	15 (27.8%)	140 (15.4%)	155 (16.0%)	**<0.001**
No	23 (42.6%)	688 (75.4%)	711 (73.6%)	
Unknown	16 (29.6%)	84 (9.2%)	100 (10.4%)	
**Loss of taste**				
Yes	16 (29.6%)	440 (48.2%)	456 (47.2%)	0.142
No	24 (44.4%)	408 (44.7%)	432 (44.7%)	
Unknown	14 (25.9%)	64 (7.0%)	78 (8.1%)	
**Altered consciousness/confusion**				
Yes	14 (25.9%)	141 (15.5%)	155 (16.0%)	**0.001**
No	23 (42.6%)	691 (75.8%)	714 (73.9%)	
Unknown	17 (31.5%)	80 (8.8%)	97 (10.0%)	
**Diarrhea**				
Yes	15 (27.8%)	260 (28.5%)	275 (28.5%)	0.397
No	25 (46.3%)	575 (63.0%)	600 (62.1%)	
Unknown	14 (25.9%)	77 (8.4%)	91 (9.4%)	
**Abdominal pain**				
Yes	12 (22.2%)	132 (14.5%)	144 (14.9%)	**0.015**
No	27 (50.0%)	698 (76.5%)	725 (75.1%)	
Unknown	15 (27.8%)	82 (9.0%)	97 (10.0%)	
**Loss of smell**				
Yes	12 (22.2%)	485 (53.2%)	497 (51.4%)	**0.001**
No	28 (51.9%)	369 (40.5%)	397 (41.1%)	
Unknown	14 (25.9%)	58 (6.4%)	72 (7.5%)	
**Vomiting/nausea**				
Yes	11 (20.4%)	134 (14.7%)	145 (15.0%)	**0.047**
No	28 (51.9%)	699 (76.6%)	727 (75.3%)	
Unknown	15 (27.8%)	79 (8.7%)	94 (9.7%)	
**Skin rash**				
Yes	5 (9.3%)	68 (7.5%)	73 (7.6%)	0.369
No	34 (63.0%)	754 (82.7%)	788 (81.6%)	
Unknown	15 (27.8%)	90 (9.9%)	105 (10.9%)	
**Cough (with hemoptysis)**				
Yes	2 (3.7%)	19 (2.1%)	21 (2.2%)	0.245
No	36 (66.7%)	782 (85.7%)	818 (84.7%)	
Unknown	16 (29.6%)	111 (12.2%)	127 (13.1%)	
**Ear pain**				
Yes	1 (1.9%)	96 (10.5%)	97 (10.0%)	0.112
No	37 (68.5%)	731 (80.2%)	768 (79.5%)	
Unknown	16 (29.6%)	85 (9.3%)	101 (10.5%)	
**Seizures**				
Yes	1 (1.9%)	3 (0.3%)	4 (0.4%)	0.162
No	36 (66.7%)	819 (89.8%)	855 (88.5%)	
Unknown	17 (31.5%)	90 (9.9%)	107 (11.1%)	
**Conjunctivitis**				
Yes	0 (0%)	28 (3.1%)	28 (2.9%)	0.630
No	38 (70.4%)	793 (87.0%)	831 (86.0%)	
Unknown	16 (29.6%)	91 (10.0%)	107 (11.1%)	

## Data Availability

The data that support the findings of this study are available on request from the corresponding author. The data are not publicly available due to privacy or ethical restrictions.

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
