# Peer review of "Characterizing Risk Factors for Hospitalization and Clinical Characteristics in a Cohort of COVID-19 Patients Enrolled in the GENCOV Study"

_viruses, 2023, doi:10.3390/v15081764_

Round 1

Reviewer 1 Report

The authors evaluated in a real-life cohort including subjects with COVID-19 the clinical features associated with hospitalization. They found that increasing continuous age and BMI, non-White/European ethnicity, hypertension, and infection by viral variants were identified as risk factors for hospitalization.

Some points should be well adressed:

1) Since the subjects enrolled are both inpatients and outpatients, it is not clear how the rate of hospitalization was assessed. Were the inpatients all admitted for COVID-related complications or due to other reasons?

2) The study period is quite long (November 2020-May 2022). This feature could impact on the outcomes of patients according to the different period. In  fact, in the first months of the COVID-19 pandemic vaccination or drugs against COVID-19 (antivirals or monoclonal antibodies) were not available and healthcare resources were stressed and unprepared.  For these reasons, I expect a higher percentage of unfavorable outcomes (hospitalization and death) in those enrolled earlier. In this regard, please add more insights in the discussion.

3) The authors reported that  the independent associations observed between increasing age and lower odds of developing specific COVID-19 symptoms were unexpected. It is likely that elderly patients who responded to the survey after Covid-19 had undersized the clinical manifestations compared to younger ones also due to a lower awareness of symptoms.

Average

Reviewer 2 Report

The overall paper is interesting and well presented. However, I would find a few biases that may eventually disturb the final statement.

A major source of potential bias in this type of study is related to losses to follow-up. Cohort members may lie, change ideas or simply refuse to accept their own health condition. In addition, the weak follow-up may ignore the in-house exposures, that again may invalidate the outcome.

A second point of concern is the vaccination. In this case, the authors should clearly confirm which vaccine, how many doses each patient received, and when...

Once we accept the fragility of these points the authors would probably need to be less assertive and sure of the results also because their final conclusions have already been discussed and presented worldwide...

Reviewer 3 Report

The paper entitled: Characterizing risk factors for hospitalization and clinical characteristics in a cohort of COVID-19 patients enrolled in the GENCOV study study is well written. In my opinion serves minor revision: shorten discussion section

Reviewer 4 Report

Important theoretical and methodological specifications must be made in order for the paper to be clearer and its argument more convincing. The objectives and the rationale of the study are not clearly stated. Research questions and hypotheses must be constructed based on more specific supporting sources, preferably as recent as possible, and clearly identifiable for each hypothesis/research question. Data gathering and data analysis can be reconsidered and discussed more comprehensively. More development and depth of the methodology and analysis are needed. The main contributions of the paper should be presented as part of the empirical discussions or critical assessment on the core research outcomes. The method is not reported in sufficient detail to allow for its replicability and reproducibility. The manuscript will benefit from further discussion of key concepts and methodological criteria in order to offer a better articulation between theory and data. You should compare your results with others in terms of concrete data for better research integrative value. The figures should be thoroughly explained. There is some discussion of the limitations of the study however these are not considered in terms of the implications on the study findings. The manuscript does not provide sufficient justification for the described and explicated findings that appear to lack empirical consistency. A Conclusions section, that should clarify the main contribution of the paper and the value added to the field, is missing. Thus, the conclusion needs to be rewritten as a distinct section so that only important results are brought out along with their interpretation, comparison with earlier studies, and implications in a more integrated fashion. The reference list does not properly follow the journal’s style.
The relationship between COVID-19 vaccine hesitancy attitudes and public health misinformation as regards patient factors which affect COVID-19 severity and outcomes has not been covered, and thus such sources can be cited:
Lăzăroiu, G., Mihăilă, R., and Braniște, L. (2021). “The Language of COVID-19 Vaccine Hesitancy and Public Health Misinformation: Distrust, Unwillingness, and Uncertainty,” Review of Contemporary Philosophy 20: 117–127. doi: 10.22381/RCP2020217.
Wells, R., Vochozka, M., and Stehel, V. (2021). “Hesitancy towards Receiving a COVID-19 Vaccine: Concerns about Side Effects and Safety, Mistrust in Government and Health Authorities, and Perceived Susceptibility and Severity of the Virus,” Review of Contemporary Philosophy 20: 163–174. doi: 10.22381/RCP20202111.
Lăzăroiu, G., Mihăilă, R., and Braniște, L. (2021). “The Language of Misinformation Literacy: COVID-19 Vaccine Hesitancy Attitudes, Behaviors, and Perceptions,” Linguistic and Philosophical Investigations 20: 85–94. doi: 10.22381/LPI2020217.

Reviewer 5 Report

This is a prospective study among patients with COVID-19 in the Greater Toronto Area in Ontario, Canada. The manuscript elegantly shows the risk factors for hospitalization, including age, BMI, non-white/European ethnicity, hypertension, and infection by viral variants. Additionally, it reveals the predictors of COVID-19 symptoms development and persistence, such as age, BMI, vaccination status, or viral strain. It could be accepted in the present form.

Round 2

Reviewer 1 Report

The authors have exhaustively answered the issues.

Good

Reviewer 4 Report

This revised version can be published.